# Outcomes of Octogenarians with Primary Malignant Cardiac Tumors: National Cancer Database Analysis

**DOI:** 10.3390/jcm11164899

**Published:** 2022-08-20

**Authors:** Mohamed Rahouma, Massimo Baudo, Anas Dabsha, Arnaldo Dimagli, Abdelrahman Mohamed, Stephanie L. Mick, Leonard Girardi, Mario Gaudino, Roberto Lorusso

**Affiliations:** 1Cardiothoracic Surgery Departments, Weill Cornell Medicine/New York Presbyterian Hospital, 525 East 68th St, Box 110, New York, NY 10065, USA; 2Surgical Oncology Department, National Cancer Institute, Cairo University, Cairo 11796, Egypt; 3Cardiac Surgery Department, SpedaliCivili di Brescia, University of Brescia, 25123 Brescia, Italy; 4Department of Cardio-Thoracic Surgery, Maastricht University Medical Centre, Maastricht University, P. Debyelaan, 25, 6202 AZ Maastricht, The Netherlands; 5Cardiovascular Research Institute Maastricht, 6202 AZ Maastricht, The Netherlands

**Keywords:** cardiac tumors, national cancer database, elderly, octogenarians

## Abstract

Data concerning age-related populations affected with primary malignant cardiac tumors (PMCTs) are still scarce. The aim of the current study was to analyze mortality differences amongst different age groups of patients with PMCTs, as reported by the National Cancer Database (NCDB). The NCDB was retrospectively reviewed for PMCTs from 2004 to 2017. The primary outcome was late mortality differences amongst different age categories (octogenarian, septuagenarian, younger age), while secondary outcomes included differences in treatment patterns and perioperative (30-day) mortality. A total of 736 patients were included, including 72 (9.8%) septuagenarians and 44 (5.98%) octogenarians. Angiosarcoma was the most prevalent PMCT. Surgery was performed in 432 (58.7%) patients (60.3%, 55.6%, and 40.9% in younger age, septuagenarian, and octogenarian, respectively, *p* = 0.04), with a corresponding 30-day mortality of 9.0% (7.0, 15.0, and 38.9% respectively, *p* < 0.001) and a median overall survival of 15.7 months (18.1, 8.7, and 4.5 months respectively). Using multivariable Cox regression, independent predictors of late mortality included octogenarian, governmental insurance, CDCC grade II/III, earlier year of diagnosis, angiosarcoma, stage III/IV, and absence of surgery/chemotherapy. With increasing age, patients presented a more significant comorbidity burden compared to younger ones and were treated more conservatively. Early and late survival outcomes progressively declined with advanced age.

## 1. Introduction

Cardiac tumors represent a heterogeneous group of masses that can potentially involve any part of the heart. The classification of cardiac neoplasms was updated in 2015 by the World Health Organization (WHO), subdividing them into benign tumors, tumor-like lesions, malignant tumors, and pericardial tumors [1].

Approximately 10% of primary cardiac tumors are malignant [2,3]. Cardiac malignancies can manifest as either primary malignant cardiac tumors (PMCTs) or as metastases from an extra-cardiac location [4,5]. Atrial myxoma is the most frequent primary cardiac tumor in adults, while rhabdomyosarcoma predominates in children [6].

PMCTs generally display aggressive biological behavior [7], and even if they are detected at an early stage and an aggressive surgical resection is performed, the survival outcome remains poor [8,9]. If surgery is not performed, results show an even more dismal prognosis, with a 10% survival rate up to 1 year following diagnosis [10,11]. The best treatment for malignant cardiac tumors consists of the combination of surgery with systemic chemotherapy [11]. Concerning perioperative chemotherapy and radiotherapy, no unanimous consensus exists regarding its efficacy [12,13,14]. Finally, for inoperable or metastatic disease, palliative chemotherapy should be offered [11].

Global life expectancy and health care have improved over the recent decades and do not seem to stop in recent projections [15]. This has led to a progressively increasing number of elderlies requiring health care and cardiac surgery in particular, including octogenarians. It is well-recognized that elderly patients undergoing cardiac surgery have a higher risk of mortality and morbidity compared to younger patients [16,17]. As a matter of fact, the Society of Thoracic Surgeons (STS) [18] score and the European System for Cardiac Operative Risk Evaluation (EuroSCORE) II [19] still include age as an incremental risk factor.

Cardiac surgery mortality and morbidities in the elderly are mostly due to the adverse cardiovascular effects of aging. In general, anemia and a suboptimal nutritional status are common [20]. Vascular dysfunction is triggered by oxidative stress and inflammation [21]. Postoperative respiratory complications are favored by reduced lung compliance, chronic lung disease, and respiratory muscle strength [22]. The glomerular filtration rate progressively falls independently of an overt pathology, and other age-associated disorders, such as diabetes and hypertension, have an additional deleterious effect on the kidney [23]. Furthermore, other chronic illnesses, such as peripheral vascular disease [24] and diabetes [25], may further negatively characterize the overall comorbidity-based profile of elderly patients [26].

Tumors may represent an additional burden of the increasingly aging population requiring health care [27,28]. However, data on the elderly population affected with PMCT are still scarce and frequently unspecified. The aim of the current study was to analyze late-mortality differences amongst different age groups of patients with PMCTs and to assess the difference in treatment pattern and perioperative mortality from a national patient population, as reported by the National Cancer Database (NCDB).

## 2. Materials and Methods

### 2.1. Study Population

The NCDB was retrospectively reviewed for PMCTs from 2004 to 2017. The NCDB is a joint project of the Commission on Cancer (CoC) of the American College of Surgeons and the American Cancer Society that includes only patients treated in the United States of America. The CoC’s NCDB and the hospitals participating in the CoC’s NCDB are the source of the de-identified data used herein; they are not responsible for the statistical validity of the data analysis or the conclusions derived by the authors. Tumors with a sequence of malignant neoplasms of more than 1 over the patient’s lifetime and those with missing survival status or time were excluded. As this study was an epidemiological study using de-identified data from the NCDB database, consent for patient participation and study publication was not required.

### 2.2. Variables, Outcomes, and Follow-Up

The following patient variables were assessed: age, sex, race, insurance status, median income quartile, metropolitan/urban/rural counties, education, great circle distance (the distance in miles between the patient’s residence and the hospital that reported the case), year of diagnosis, Charlson–Deyo comorbidity condition index (CDCC, categorized as (0 or 1 vs 2 or 3)), histology, tumor size, analytic stage, facility type, different treatment patterns, such as surgery, radiotherapy, and chemotherapy, and 30-day mortality and late mortality. Facility type was categorized as 1. academic/integrated facilities (including academic/research programs and integrated network cancer programs) or 2. community facilities (including community cancer programs and comprehensive community cancer program).

The primary outcome of this study was late mortality differences amongst different age categories (octogenarian, septuagenarian, and younger-age group), while secondary outcomes included differences in treatment patterns and early (30-day) mortality, as well as its annual proportion among different age categories throughout the study period. Furthermore, sensitivity analysis was conducted on surgical patients.

### 2.3. Statistical Analysis

Categorical variables were presented as frequency counts and percentages and compared between groups using a Chi-square test. After checking normality of continuous variables, they were presented as means and standard deviations if normally distributed and were compared between groups using analysis of variance (ANOVA). If they were not normally distributed, median and interquartile ranges were presented and compared between groups using the Kruskal–Wallis test. All tests were 2-sided, and the alpha level was set at 0.05 for statistical significance.

Early (30-day) mortality predictors were identified through univariate and multivariable logistic regression among the surgical subgroups. Late all-cause mortality was assessed using Kaplan–Meier curves and compared between the groups using the log-rank test. Median follow-up time was assessed among different age categories utilizing reversed Kaplan–Meier curves. A univariate Cox proportional hazards model was used to assess hazard ratios of late all-cause mortality amongst different age categories (octogenarian, septuagenarian and younger-age group). The proportional hazards assumption was tested using Schoenfeld residuals. Independent predictors of late mortality were assessed using multivariate Cox regression.

All statistical analyses were performed using R version 4.1.1 within RStudio with the help of the following packages “tableone”, “Publish”, “Hmisc”, and “survminer” [29,30,31].

## 3. Results

### 3.1. Demographics

Out of the 110,991 malignant soft-tissue tumors present in the NCDB, 907 malignant cardiac tumors were identified. Due to the presence of a sequence of malignant neoplasms of more than one over the lifetime of the patient, 107 patients were excluded. In addition, 64 additional patients were excluded due to missing survival time or status. Therefore, a total of 736 patients were finally included in the study analysis.

Median age was 52 years (interquartile range (IQR): 37 to 65 years) and included 72 septuagenarians and 44 octogenarians. Overall, there were 352 females (47.8%). Almost 80% of the study population was white, followed by black (13.6%) and other (6.5%). Less than 10% were uninsured/unknown, while 56% had private insurance and 34.2% governmental insurance (Table 1).

The three age groups showed some significant differences: among the different age groups, white was the most prevalent race, while black was the second most prevalent for younger patients and the least prevalent in septuagenarians and octogenarians (*p* = 0.007). Younger patients were more likely to have private insurance compared to elderly patients, who showed a greater predominance of governmental insurance (*p* < 0.001). Finally, elderly patients showed a significantly higher CDCC degree when compared to the younger group (*p* = 0.002).

### 3.2. Histology

Among all patients, angiosarcoma was the most common histology (43.8%), followed by fibrosarcoma and leiomyosarcoma (5.2% each), while Ewing sarcoma (0.3%) was the least prevalent one. This trend was maintained in the younger group. Although angiosarcoma was still the most prevalent cardiac tumor in the elderly, the second most common was different between septuagenarians and octogenarians (Table 1).

There were no significant differences between the age groups as far as grade and tumor stage were concerned (*p* = 0.732 and *p* = 0.34, respectively).

### 3.3. Treatment

A total of 432 patients (58.7%) underwent surgery; 374 (50.8%) received chemotherapy, and 130 (17.7%) received radiotherapy. Treatment analysis showed significant differences between age groups as far as surgery and chemotherapy were concerned (*p* = 0.037 and *p* < 0.001, respectively): with increasing age, surgery and chemotherapy were performed less frequently. Amongst the surgically treated patients, 219 (50.7%) patients had chemotherapy and 73 (16.9%) had radiation therapy (Table 1). Also in this subgroup, septuagenarians and octogenarians received less chemotherapy than younger patients (*p* < 0.001) (Appendix A).

### 3.4. Survival Analysis

Overall, the 30-day mortality was 8.7% (6.7%, 14.0%, and 36.8% for younger, septuagenarians, and octogenarians, respectively, *p* < 0.001) (Table 1). Over the years, the direct relation between age and mortality was preserved (Appendix A). In the surgically treated patients, 30-day mortality was 9.0% (7.0%, 15.0%, and 38.9% for younger, septuagenarians, and octogenarians, respectively, *p* < 0.001) (Appendix A).

Median follow-up was 42.3 months in the octogenarian category vs. 61.4 and 67 months in the septuagenarian and younger-age categories, respectively. Median overall survival was 12.98 (95%CI: 11.50–14.65), 6.03 (95%CI: 2.60–9.30), and 1.33 (95%CI: 0.49–4.14) for younger, septuagenarians, and octogenarians respectively.

Two-year overall survival was 31.40%, 20.48%, and 11.36% for younger, septuagenarians, and octogenarians, respectively. Five-year overall survival was 14.10%, 9.78%, and 6.82% for younger, septuagenarians, and octogenarians, respectively (*p* < 0.001, Figure 1).

Using multivariable Cox regression, independent predictors of late mortality among the entire cohort included octogenarian (hazard ratio (HR): 1.810, 95% confidence interval (CI): [1.277;2.565], *p* < 0.001, younger set as reference), governmental insurance (HR: 1.460, 95%CI: [1.071;1.992], *p* = 0.017, no/unknown insurance set as reference), CDCC grade II/III (HR: 2.000, 95%CI: [1.514;2.642], *p* < 0.001, CDCC grade 0/I set as reference), year of diagnosis (HR: 0.960, 95%CI: [0.939;0.982], *p* < 0.001), angiosarcoma (HR: 1.295, 95%CI: [1.092;1.536], *p* = 0.003, other histology set as reference), stage III/IV (HR: 1.298, 95%CI: [1.082;1.556], *p* = 0.005, stage 0/I/x set as reference), surgery (HR: 0.478, 95%CI: [0.400;0.572], *p* < 0.001), and chemotherapy (HR: 0.601, 95%CI: [0.501;0.721], *p* < 0.001) (Table 2).

Using multivariable logistic regression, independent predictors of 30-day mortality among the surgical cohort included octogenarian (odds ratio (OR): 7.31, 95%CI: [2.55;20.94], *p* < 0.01, younger set as reference) and CDCC grade II/III (HR: 2.46, 95%CI: [1.03;5.86], *p* = 0.042, CDCC grade 0/I set as reference) (Appendix A).

## 4. Discussion

Age is currently one of the most commonly reported risk factors for adverse outcomes in cardiac surgery [16,17] and the most frequently cited reason for not operating on otherwise eligible patients [32,33]. Nevertheless, a clear definition of the “elderly” patient in cardiac surgery has never been established. The cut-off used in published articles ranges from 65 to 80 years [34]. The present study evaluated the outcome of 736 patients with PMCTs from the NCDB based on three age groups (younger, septuagenarians, and octogenarians).

The baseline characteristics significantly differed between the age groups for race, insurance status, comorbidity index (CDCC), histology, and cancer stage. As far as patients’ sex was concerned, a trend of higher female prevalence was noted for increasing age (*p* = 0.064). Previous studies on PMCTs showed that females were significantly older and had a lower cancer stage in comparison to males [35].

Treatment strategies significantly differed among the age groups: with increasing age, less surgery (*p* = 0.037) and less chemotherapy (*p* < 0.001) were chosen. Early survival outcomes were significantly impacted by age. Higher 30-day mortality increased with age in the whole population (*p* < 0.001) and in the surgical group (*p* < 0.001). The same trend was noted for long-term death from any cause in the overall cohort (log-rank *p* < 0.001) and in the surgically treated (log-rank *p* = 0.004). Independent predictors of late mortality among the entire cohort included octogenarian age group, governmental insurance, advanced CDCC grade, angiosarcoma histology, and stage III/IV. Surgery and chemotherapy were associated with longer survival benefits. Notably, radiotherapy was not an independent predictor of survival.

Generally, in cardiac surgery, the chronological age may not be per se a risk factor for higher perioperative and postoperative morbidity and mortality. The more crucial factor is the biological age, particularly the associated comorbidities and the timing of the surgical procedure in the course of the disease. In a meta-analysis by Wiegmann et al. [36], the authors reported how elderly patients, even those over 80 years old, can benefit in all aspects of cardiac surgery, as long as individually adapted operative techniques are considered. However, for PMCTs, the neoplastic nature of the disease further complicates the clinical picture, and such statements may not be completely applicable. In the present analysis, despite surgery and chemotherapy being independent predictors of longer survival, elderly patients were at higher risk of early and late mortality. As a result, with these premises and the presence of limited benefit in the face of high early mortality among octogenarians, it is appropriate to convey such data to the patients and their families in order to obtain a shared decision on the treatment plan among the available therapeutic options.

Previous studies on PMCTs showed that a survival disparity among patients with different insurance status exists [37]. This was confirmed in the current analysis: governmental insurance was associated with a significant increase in late mortality with multivariable regression (HR: 1.460, 95%CI: [1.071;1.992], *p* = 0.017, no/unknown insurance set as reference). Furthermore, uninsured and governmental-insured patients were shown to be associated with higher early-mortality rates after cardiac valve surgery in a recent paper by Hoyler [38]. While the current analysis showed that patients of advanced age had a lower prevalence of advanced cancer stages (*p* = 0.037), it also confirmed that advanced age was associated with higher comorbidities (CDCC, *p* = 0.002). The prior literature [39] has consistently shown that the mortality rate is higher in elderly patients undergoing cardiac surgery, and that is reiterated in this study. Nevertheless, multiple studies [40,41,42] reported that better perioperative mortality and morbidity can be achieved in this subset of patients than is expected. This is likely due to multiple reasons, including better preoperative selection of patients, improvement in perioperative care, and the ability to rescue patients from postoperative complications. Moreover, currently, less invasive treatment options may recruit patients referred for cardiac surgery who could not have been considered in the past [43]. In general, older patients often require longer postoperative stays and therefore an increase in resources compared to younger patients [44]. Despite the high rates of early re-hospitalization, octogenarians with prolonged intensive care unit length of stay have been reported to have acceptable functional survival at 1 year [45].

Multidisciplinary discussions are of paramount importance for older patients, who often have issues of frailty and multiple comorbidities to consider, in order to establish the best treatment option for each individual.

### Study Limitations

The retrospective nature of the present study may have introduced important biases and confounding factors that cannot be avoided even after different types of statistical adjustments. In addition, classifying some categorical variables as “unknown” is a probable source of bias. It was not possible to perform a complete subgroup analysis of multimodal therapy due to the small sample size of some groups.

The data obtained from the NCDB are subject to human error in both coding and data input due to the multiple registrars. Furthermore, no genetic data are provided in the NCDB. Moreover, specific chemotherapy regimens were not included, and the diagnostic imaging modalities, clinical presentation, and tumor location within the heart were not available. However, diagnoses were confirmed with histological examinations after resection or biopsy.

## 5. Conclusions

The most frequent PMCT is angiosarcoma at all considered ages. With increasing age, patients presented a more significant comorbidity burden compared to younger ones and were treated more conservatively. Early and late survival outcomes progressively declined with advanced age. However, most recent data show that acceptable results might be achieved with the use of surgery and chemotherapy, although elderly patients are often not considered appropriate candidates for such therapies. Finally, independent predictors of late mortality among the entire cohort included octogenarian age group, governmental insurance, advanced CDCC grade, angiosarcoma histology, and stage III/IV. On the contrary, surgery and chemotherapy were associated with longer survival benefits.

## Figures and Tables

**Figure 1 jcm-11-04899-f001:**
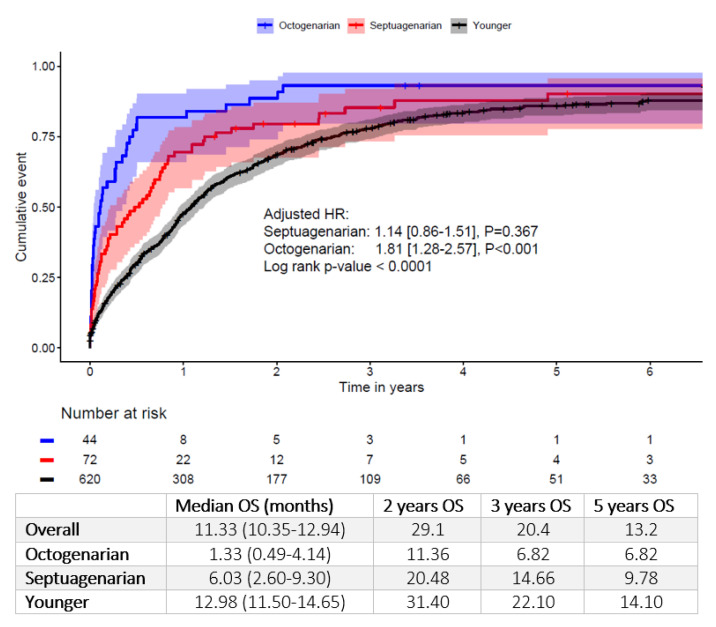
Cumulative death from any cause among different age categories (the younger category was used as reference for the obtained hazard ratios).

**Table 1 jcm-11-04899-t001:** Criteria of the included patients.

	Overall	Younger	Septuagenarian	Octogenarian	*p*
* **n** *	736	620	72	44	
**Age (median [IQR])**	52.00 [37.00, 65.00]	47.00 [35.00, 58.00]	74.00 [72.00, 77.00]	84.00 [82.00, 86.00]	<0.001
**Female (%)**	352 (47.8)	285 (46.0)	41 (56.9)	26 (59.1)	0.064
**Race (%)**					0.007
• **White**	575 (79.9)	474 (78.1)	64 (90.1)	37 (88.1)	
• **Black**	98 (13.6)	95 (15.7)	2 (2.8)	1 (2.4)	
• **Others**	47 (6.5)	38 (6.3)	(7.0)	4 (9.5)	
**Insurance status**					<0.001
• **No insurance/Unknown**	72 (9.8)	69 (11.1)	3 (4.2)	0 (0.0)	
• **Private**	412 (56.0)	396 (63.9)	9 (12.5)	7 (15.9)	
• **Governmental**	252 (34.2)	155 (25.0)	60 (83.3)	37 (84.1)	
**Median income quartile (%)**					0.136
• **Less than USD 40,227**	124 (17.7)	110 (18.7)	8 (11.1)	6 (14.0)	
• **USD 40,227– USD 50,353**	145 (20.7)	116 (19.8)	14 (19.4)	15 (34.9)	
• **USD 50,354– USD 63,332**	170 (24.2)	143 (24.4)	21 (29.2)	6 (14.0)	
• **USD 63,333 or more**	263 (37.5)	218 (37.1)	40.3	16 (37.2)	
**Urban/Rural counties (%)**					0.62
• **Metropolitan**	591 (84.3)	495 (84.0)	59 (84.3)	37 (88.1)	
• **Urban**	98 (14.0)	82 (13.9)	11 (15.7)	5 (11.9)	
• **Rural**	12 (1.7)	12 (2.0)	0 (0.0)	0 (0.0)	
**No high school graduate quartile (%) ¶**					0.053
• **17.6% or more**	149 (21.1)	134 (22.8)	10 (13.9)	5 (11.4)	
• **10.9–17.5%**	174 (24.7)	141 (23.9)	15 (20.8)	18 (40.9)	
• **6.3–10.8%**	194 (27.5)	162 (27.5)	23 (31.9)	9 (20.5)	
• **Less than 6.3%**	188 (26.7)	152 (25.8)	24 (33.3)	12 (27.3)	
**Great circle distance (miles; median [IQR]) ¶¶**	17.00 [6.20, 45.80]	17.80 [6.40, 48.85]	15.00 [7.27, 29.80]	10.30 [3.98, 33.22]	0.069
**Facility type (%)**					<0.001
• **Academic/integrated**	344 (46.7)	282 (45.5)	39 (54.2)	23 (52.3)	
• **Community**	184 (25.0)	130 (21.0)	33 (45.8)	21 (47.7)	
• **Unknown**	208 (28.3)	208 (33.5)	0 (0.0)	0 (0.0)	
**CDCC (0 or 1/2 or 3) (%) ***	671/65 (91.2/8.8)	575/45 (92.7/7.3)	59/13 (81.9/18.1)	37/7 (84.1/15.9)	0.002
**Year of diagnosis (median [IQR])**	2011.00 [2007.00, 2014.00]	2011.00 [2007.00, 2014.00]	2012.00 [2008.75, 2014.00]	2012.00 [2008.00, 2014.25]	0.125
**Histology (%)**					<0.001
• **Angiosarcoma**	322 (43.8)	289 (46.6)	24 (33.3)	9 (20.5)	
• **Ewing sarcoma**	2 (0.3)	2 (0.3)	0 (0.0)	0 (0.0)	
• **Fibrosarcoma**	38 (5.2)	33 (5.3)	2 (2.8)	3 (6.8)	
• **Giant cell sarcoma**	34 (4.6)	25 (4.0)	7 (9.7)	2 (4.5)	
• **Leiomyosarcoma**	38 (5.2)	31 (5.0)	4 (5.6)	3 (6.8)	
• **Liposarcoma**	6 (0.8)	3 (0.5)	3 (4.2)	0 (0.0)	
• **Malignant fibrous histiocytoma (MFH)**	13 (1.8)	13 (2.1)	0 (0.0)	0 (0.0)	
• **MPNST ***	4 (0.5)	4 (0.6)	0 (0.0)	0 (0.0)	
• **Myxosarcoma**	23 (3.1)	22 (3.5)	1 (1.4)	0 (0.0)	
• **Osteosarcoma**	18 (2.4)	18 (2.9)	0 (0.0)	0 (0.0)	
• **Others/Unclassified**	186 (25.3)	140 (22.6)	22 (30.6)	24 (54.5)	
• **Rhabdomyosarcoma (RMS)**	21 (2.9)	12 (1.9)	6 (8.3)	3 (6.8)	
• **Synovial sarcoma**	31 (4.2)	28 (4.5)	3 (4.2)	0 (0.0)	
**Grade (poorly differentiated/anaplastic) (%)**	362 (49.2)	304 (49.0)	38 (52.8)	20 (45.5)	0.732
**Tumor size (in mm; median [IQR])**	60.00 [44.00, 80.00]	60.00 [45.00, 80.00]	60.00 [41.25, 73.75]	50.00 [40.00, 70.50]	0.34
**Analytic stage group (%)**					0.037
• **Stage 0/I/x**	355 (48.2)	293 (47.3)	37 (51.4)	25 (56.8)	
• **Stage II**	89 (12.1)	70 (11.3)	9 (12.5)	10 (22.7)	
• **Stage III/IV**	292 (39.7)	257 (41.5)	36.1	9 (20.5)	
**Surgery (%)**					0.037
• **No**	301 (40.9)	244 (39.4)	32 (44.4)	25 (56.8)	
• **Yes**	432 (58.7)	374 (60.3)	40 (55.6)	18 (40.9)	
• **Unknown**	3 (0.4)	2 (0.3)	(0.0)	1 (2.3)	
**Radiation (%)**					0.148
• **No**	581 (78.9)	485 (78.2)	57 (79.2)	39 (88.6)	
• **Yes**	130 (17.7)	115 (18.5)	13 (18.1)	2 (4.5)	
• **Unknown**	25 (3.4)	20 (3.2)	(2.8)	3 (6.8)	
**Chemotherapy (%)**					<0.001
• **No**	308 (41.8)	229 (36.9)	44 (61.1)	35 (79.5)	
• **Yes**	374 (50.8)	353 (56.9)	20 (27.8)	1 (2.3)	
• **Unknown**	54 (7.3)	38 (6.1)	8 (11.1)	8 (18.2)	
**30-day mortality (Alive/Dead) (%)**	410/39 (91.3/8.7)	361/26 (93.3/6.7)	37/6 (86.0/14.0)	12/7 (63.2/36.8)	<0.001

¶ This item provides a measure of the number of adults aged 25 or older in the zip code of patients who did not graduate from high school, and is categorized as equally proportioned quartiles among all US zip codes. ¶¶ The “great circle” distance in miles between the patient’s residence and the hospital that reported the case. * CDCC: Charlson–Deyo comorbidity condition, MPNST: malignant peripheral nerve sheath tumors, USD: United States Dollar.

**Table 2 jcm-11-04899-t002:** Independent predictors of late mortality among the entire cohort.

Variable	Units	Univariate Analysis Hazard Ratio (95% CI) p-Value	Multivariable Analysis Hazard Ratio (95% CI) p-Value
**Age**	Younger	Ref	Ref
	Septuagenarian	**1.507 [1.159;1.958] 0.002**	1.139 [0.858;1.512] 0.367
	Octogenarian	**2.569 [1.872;3.526] <0.001**	**1.810 [1.277;2.565] <0.001**
**Sex**	Male	Ref	
	Female	1.031 [0.880;1.207] 0.708	
**Race**	White	Ref	
	Black	1.071 [0.848;1.352] 0.565	
	Others	1.305 [0.933;1.825] 0.120	
**Insurance**	No/Unknown	Ref	Ref
	Private	0.862 [0.649;1.144] 0.3037	0.962 [0.721;1.283] 0.790
	Governmental	**1.451 [1.081;1.947] 0.013**	**1.460 [1.071;1.992] 0.017**
**Median income quartile**	Less than USD 40,227	Ref	
	Higher than USD 40,227	1.093 [0.882;1.353] 0.417	
**Urban/Metropolitan**	Urban	Ref	
	Metropolitan	0.965 [0.763;1.220] 0.764	
	Rural	1.077 [0.592;1.958] 0.808	
**CDCC**	0/I	Ref	Ref
	II/III	**1.614 [1.236;2.108] <0.001**	**2.000 [1.514;2.642] <0.001**
**Year of diagnosis**		0.989 [0.968;1.010] 0.308	**0.960 [0.939;0.982] <0.001**
**Histology**	Others	Ref	Ref
	Angiosarcoma	**1.312 [1.118;1.539] <0.001**	**1.295 [1.092;1.536] 0.003**
**Analytic stage**	Stage 0/I/x	Ref	Ref
	Stage II	0.862 [0.666;1.116] 0.261	0.936 [0.715;1.224] 0.628
	Stage III/IV	1.164 [0.983;1.379] 0.078	**1.298 [1.082;1.556] 0.005**
**Facility type**	Community	Ref	
	Academic/Integrated	1.092 [0.900;1.326] 0.373	
**Surgery**	No	Ref	Ref
	Yes	**0.494 [0.419;0.581] <0.001**	**0.478 [0.400;0.572] <0.001**
**Radiation**	No	Ref	
	Yes	0.833 [0.677;1.025] 0.085	
**Chemotherapy**	No	Ref	Ref
	Yes	**0.682 [0.577;0.806] <0.001**	**0.601 [0.501;0.721] <0.001**

Variables with *p* < 0.05 in univariate analysis were included in multivariable analysis.

## Data Availability

This is a national cancer database that we got approval to use, but it is not publically available.

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
