# Peer review of "Outcomes of Octogenarians with Primary Malignant Cardiac Tumors: National Cancer Database Analysis"

_jcm, 2022, doi:10.3390/jcm11164899_

Round 1

Reviewer 1 Report

This article is well written and informative both for the doctors treating primary malignant cardiac tumors and for the patients being treated. As the authors indicated, the data presented here are of importance in the decision of the treatment plan for the elderly. There seems to be no important revisions needed in the present manuscript.

Just one additional explanation is needed for the international readers. Does the NCDB data include only the patients treated in the United States of America? Please mention about it in the 2.1. Study Population.

Thank you for presenting this important study.

Author Response

Comment 1: This article is well written and informative both for the doctors treating primary malignant cardiac tumors and for the patients being treated. As the authors indicated, the data presented here are of importance in the decision of the treatment plan for the elderly. There seems to be no important revisions needed in the present manuscript.

Answer 1: We thank the Reviewer for having taken the time to read and comment upon our manuscript. We value her/his suggestions.

Comment 2: Just one additional explanation is needed for the international readers. Does the NCDB data include only the patients treated in the United States of America? Please mention about it in the 2.1. Study Population.

Answer 2: We appreciate the Reviewer’s suggestion and specified the origin of patient data.

Changes 2: The NCDB is a joint project of the Commission on Cancer (CoC) of the American College of Surgeons and the American Cancer Society that include only patients treated in the United States of America.

Sherif Khairallah, smk4005@med.cornell.edu, was added as co-author

Reviewer 2 Report

Mohamed Rahouma et al. present a retrospective study on primary cardiac cancer on data collected from the national cancer database. They focused their study on the impact that age has on mortality. On a final sample of 736 patients, characteristics are clearly shown as the results of the multivariable Cox regression. The results were that older patients are more likely to die and older age group, governmental insurance, advanced CDCC grade, angiosarcoma histology, and stage III/IV are independent risk factor for late mortality. 

The study is well conducted and well presented, even if the results and the conclusione are not ground-breaking. Nevertheless, the topic is not very commonly investigated and the sample size is bigger than most of published literature, therefore I find it very interesting. I would suggest to add a comment a bit more on the role of insurance (line 233). 

Author Response

Comment 1: Mohamed Rahouma et al. present a retrospective study on primary cardiac cancer on data collected from the national cancer database. They focused their study on the impact that age has on mortality. On a final sample of 736 patients, characteristics are clearly shown as the results of the multivariable Cox regression. The results were that older patients are more likely to die and older age group, governmental insurance, advanced CDCC grade, angiosarcoma histology, and stage III/IV are independent risk factor for late mortality.

Answer 1: We thank the Reviewer for having taken the time to read and comment upon our manuscript. We value her/his suggestions.

Comment 2: The study is well conducted and well presented, even if the results and the conclusion are not ground-breaking. Nevertheless, the topic is not very commonly investigated and the sample size is bigger than most of published literature, therefore I find it very interesting. I would suggest to add a comment a bit more on the role of insurance (line 233).

Answer 2: We appreciate the Reviewer’s suggestion and added more details on the role of insurance.

Changes 2: Previous studies on PMCTs showed that a survival disparity among patients with different insurance status exists[37]. This was confirmed in the current analysis: governmental insurance was associated with a significant increase in late mortality at multivariable regression (HR: 1.460, 95%CI: [1.071;1.992], p=0.017, no/unknown insurance set as reference). Besides, uninsured and governmental insured patients were shown to be associated to higher early mortality rates after cardiac valve surgery in a recent paper by Hoyler[38].

The following reference was added:

  1. Hoyler MM, Feng TR, Ma X, Rong LQ, Avgerinos DV, Tam CW, et al. Insurance Status and Socioeconomic Factors Affect Early Mortality After Cardiac Valve Surgery. J Cardiothorac Vasc Anesth 2020;34:3234-42. https://doi.org/10.1053/j.jvca.2020.03.044

Sherif Khairallah, smk4005@med.cornell.edu, was added as co-author